# Developing the Framework of Drone Curriculum to Educate the Drone Beginners in the Korean Construction Industry

**Seojin Moon and Jongho Ock ***

Department of Architectural Engineering, Seoul National University of Science & Technology, Seoul 139-743, Republic of Korea; msd1521@gmail.com
* Correspondence: ockjh@seoultech.ac.kr; Tel.: +82-10-7701-0504

**Abstract:** Both drones and laser scanners digitally take the as-built context of an object into the computer, and the data taken are transmitted to a Building Information Modeling (BIM) world to create accurate 3D models. Although the laser scanner is the leading method of the Scan-to-BIM procedure, many professionals indicate drawbacks of the technology and point out the drone is an alternative that can improve the shortcomings, leading to the UAV-to-BIM process. The Korean construction industry plans to implement drone technology for scrutinizing as-built construction quality by 2025. However, drones are not popular in construction projects. Korean universities where Construction Engineering and Management programs have been implemented have requested to develop a drone curriculum for construction professionals. Since the majority of the professionals are not familiar with drone operation, in order for the schools to be successful in developing the curriculum, it is very necessary to perform a preliminary experimental study for identifying the essential education contents that are appropriate for drone beginners. The main objective of this paper is to perform a study for drone beginners and recognize the recommendations and the framework of a drone curriculum that will be beneficial for the schools to develop a comprehensive curriculum later on.

**Keywords:** drone; laser scanning; drone curriculum; construction operation monitoring; Smart Construction; Construction 4.0





## 1. Introduction

### 1.1. Research Background and Objective

The construction industry worldwide has been notorious for its negative features, i.e., numerous accidents and lower productivity and digitalization levels than other industries such as manufacturing and medicine. According to a McKinsey study, the industry suffers from a great deal of inefficiency; so to speak, large construction projects usually take 20% longer than planned to complete and are up to 80% over budget [1,2].

The chronic problems are dedicated to the unique nature of construction operations including being labor-intensive, complex, and fragmented. To take care of the problems, many developed countries have enhanced innovative construction technologies in combination with a variety of digital technologies. This phenomenon has been referred to as Smart Construction [3,4] or Construction 4.0 [5–7].

Among the technologies, unmanned aerial vehicles (UAVs), known as a drones, have been recently well acknowledged as a significant technique for collecting valuable data in diverse construction processes. DroneDeploy, a cloud software platform for commercial drones, presented remarkable statistics on drone applications in 2018 [8]. It pointed out that drone use in the construction industry has risen steeply up to 239% compared to 2017, which is much greater than other industries. According to Allied Market Research, the global market size of construction drones was valued at USD 4800 million in 2019 and is estimated to move up to USD 11,970 million by 2027 [9].

A drone does a similar task to laser scanning (LS) in monitoring as-built construction quality through a Scan-to-BIM process [10,11]. Both technologies transfer the as-built conditions of an object into the computer world as a 3D BIM model and make the BIM model be compared with the as-planned data of the object. Additionally, by means of the Scan-to-BIM context, the absent design data, e.g., 2D drawings, can be produced through Reverse Engineering (RE). The RE processes allow engineers to come up with how a piece of a building has been created so that they can rebuild the product [12].

While the LS has been told the major technique of the Scan-to-BIM process, many digital professionals address the shortcomings of the LS such as being a time-consuming process, the static nature in collecting data, and casting a shadow due to the scanning mechanism [13] and emphasize that drones are an alternative which readily recovers the problems of the LS. A drone is well employed when capturing the facades of facilities due to its speed. Different from laser scanners, the drone can scan rooftops. Owing to its mobile character, the drone undergoes much less shadowing from adjacent objects. Drone technology delivers the precise point clouds that are to be processed to BIM software such as Revit and Archicad to generate a 3D model. It is the reason why UAV-to-BIM is referred to interchangeably with the Scan-to-BIM [14].

In 2018, the Ministry of Land, Infrastructure and Transport, Korea (MOLIT), the main governmental organization in charge of managing public policies to provide quality infrastructures in Korea, established the "Smart Construction Technology Roadmap" as the policy of implementing Construction 4.0 and announced the plan to adopt drones for enhancing the as-built quality of major public projects by 2025 [4]. It is necessary for Korean contractors to train their engineers to be competitive in operating the drone. However, the drone is not so popular in construction projects in Korea. Among 12,651 general contractors nationwide, only the top 50 level contractors have from time to time utilized the technology in their projects [15].

A successful drone flight is not just operating the equipment but gathering the image data and editing the data to fit well to the objective of the intended operation. The tasks require skills and experience. The universities where Construction Engineering and Management (CE&M) programs have been implemented in Korea have been requested to develop the curriculum for providing drone education for the engineers. While a number of the schools deliver either graduate or undergraduate BIM curricula, they have scarcely operated the drone or the LS curriculum. The CONECTtech Lab at Georgia Tech, Georgia, USA, has recently carried out a course called "Technology Applications in the Construction Industry" [16]. he course encompasses multiple construction technologies, i.e., the LS, 4D BIM, photogrammetry, virtual reality, mobile applications, and project management software. However, the major concentration and attraction for students of the course are positioned on drones. The CONECTtech Lab provides a good reference for Korean universities to develop a drone-oriented curriculum in the CE&M program.

There are very few drone experts in Korea knowledgeable enough to develop the curriculum. Developing the curriculum should systematically organize clear learning objectives, proper methodologies to the objectives, potential limitations, the amount of time required, and the final product of education [17,18]. Each component affects and interacts with others. To avoid trials and errors in progressing the curriculum and successfully implementing it later on, it is very essential for the schools to implement a preliminary study based on experimental drone application so as to specifically identify the necessary education for drone beginners and the potential risks that lie hid there. On the basis of the findings from this study, a comprehensive drone curriculum for the engineers can be initialized in the end.

The main objective of this study is to perform the preliminary study through experimental drone application with drone beginners in the Korean construction industry. This study empirically identifies the pros and cons of the drone in monitoring as-built conditions of a facility and the needs for collaboration of the drone with the LS technology. Based on this study's result, the recommendations are identified that would be beneficial

to promote the comprehensive drone curriculum to support the engineers to enhance the as-built quality of facilities compared to their as-designed data. In the process of identifying the recommendations, the following questions are specifically answered: (1) Is it necessary to develop the drone education curriculum in universities for construction engineers? (2) What contents are essentially embraced in the curriculum? (3) What formats and supports are desirable to successfully develop and implement the curriculum?

### 1.2. Research Methodology

The research objective was met through following the five activities: (1) reviewing the existing body of knowledge on a drone and a laser scanner in construction, (2) preparing for the experimental operation of the technologies, i.e., participants, equipment to be used, facilities to be studied, and professionals who can support the experiment, (3) implementing the operation of the equipment around the facilities, (4) presenting the findings from the experiment and discussing needs for education, and (5) identifying the recommendations valuable for the schools to exploit a comprehensive drone curriculum. Figure 1 shows the sequential procedures of the study.

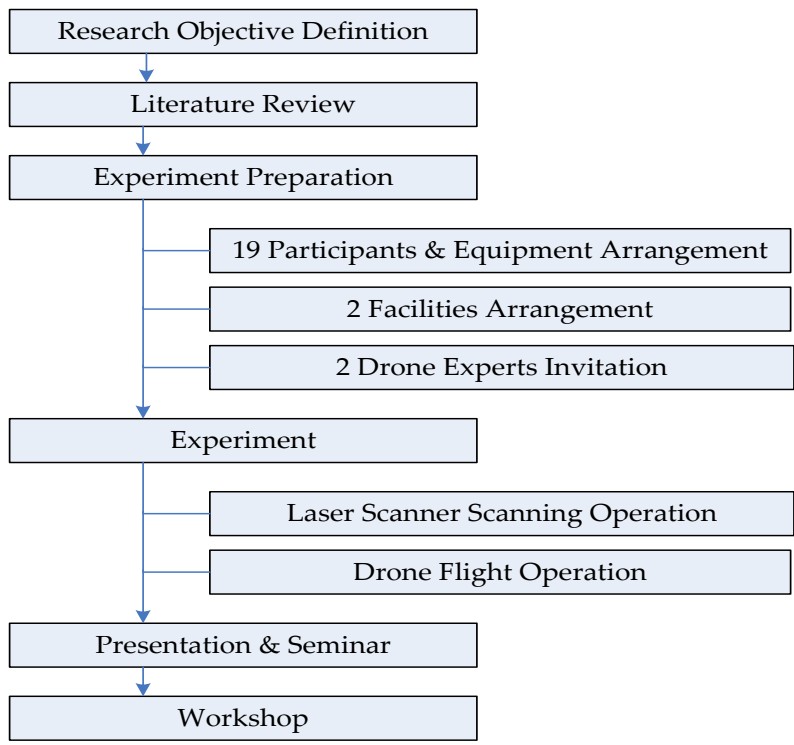

**Figure 1.** Research procedures of the study.

In step (2), the authors arranged 19 students in the Free-form Facade Engineering (FFE) class as the participants of the experiment, which is one of the existing CE&M classes at Seoul National University of Science and Technology (SNUST) in Seoul, Republic of Korea. The class has used LS technology for several years to educate students on how to measure the construction quality of the free-form building facades. The authors incorporated a drone education module into the class to test the usability of the drone and to compare the distinctions of the two technologies in capturing as-built conditions of a facility. Twelve out of the nineteen were undergraduates, and the other seven were part-time graduates. Table 1 summarizes the statistics regarding the 19 participants of the experiment.

**Table 1.** Demographic information of the experiment participants.

| Gender | | Age | | Construction Job Experience | | Drone Familiarity | |
|---|---|---|---|---|---|---|---|
| Male | 13 (68%) | 21–30 | 12 (63%) | 5 yrs or Less | 12 (63%) | Yes | 0 (0%) |
| Female | 4 (32%) | 31–40 | 4 (21%) | 6–10 yrs | 4 (21%) | No | 19 (100%) |
| | | 41–50 | 3 (15%) | 11–20 yrs | 3 (15%) | | |

Two drone training professionals were invited to guide the experimental activities of the participants from a private drone flight training institution. The role of the professionals featured supporting the authors to manage experimental procedures overall and participating in the workshop after the experiment. Additionally, two facilities with different shapes and complexities in the SNUST campus were arranged as the subject facilities to be experimented.

In step (4), after finishing step (3), a presentation seminar was implemented. The participants presented their experimental findings, problems, feelings, and any difficulties freely in the seminar. The authors and the two professionals participated in the seminar to discuss and critique the findings and deliver their experience regarding those items.

In step (5), the workshop having a focus-group interview (FGI) format was conducted with the two professionals and the seven part-time graduates to specifically determine the recommendations beneficial for developing the drone curriculum on the basis of the results in step (4). FGIs are interviews operated with a group of participants to build up diverse information [19]. The authors performed the workshop instead of a questionnaire survey since the former can obtain more specific and comprehensive findings.

## 2. Literature Review

### 2.1. Industry 4.0 and Construction 4.0

Industry 4.0 has led distinguished technological advancements worldwide for the last 10 years. The German Federal Government (GFG) defined in 2011 Industry 4.0 as "a new technological age for manufacturing that uses cyber-physical systems and IoT to connect production technologies with smart production processes [5]". Schwab [20] later described it as the phase that enables the rich combination of people with machines by means of the aid of information technology. Additionally, Helman et al. [21] explained it as a new model of value chain throughout the lifecycle of merchandises.

Construction 4.0 is the term used to represent Industry 4.0 in the construction industry [22]. The first article that mentioned Industry 4.0 along with construction was shown in September 2014 [23]. Regarding Construction 4.0, Ronald Berger initially described it in his article in 2016 [24]. The European Industry Construction Federation (FIEC) described it as the representation of Industry 4.0 in the construction industry to effectively generate and manage a variety of built facilities through digitalization [25].

### 2.2. Drone and LS Application in Construction

LS digitally takes the as-built configurations of physical objects into the computer world by a line of laser light [26]. The drone performs a function similar to the laser scanner. Both of them have become an important method necessary for meeting the Scan to BIM cycle [10,11]. Relying on the large amounts of data from the two technologies, architects and project managers can approach strong understandings that allow for having a very practical model of the project. This is crucial to obtaining informed and data-driven decisions and significant for obtaining as-is circumstances and crafting subsequent design changes. In the Scan-to-BIM procedure, the image occupied by the two technologies is conveyed into a 3D context by using software which shifts the scanned image to the BIM world for the purpose of producing precise as-built models [27].

By means of the Scan-to-BIM context, the overlooked design data, e.g., 2D drawings or 3D models, are made through Reverse Engineering (RE). RE is frequently referred to as back

engineering or reverse design and is the procedure where diverse products are disintegrated to mine design intelligence, codes, and information from them [6]. Engineering generally creates drawings of the product to be manufactured based on engineering translation, and the product is manufactured on the drawings. However, in the domain of the RE, the product drawings are mined out from the finished product. The RE process allows for coming up with how parts of a product are originated, so that makes it possible to rebuild it under the circumstance where the drawings are not available.

Ock [11] presents the RE process through a case study as shown in Figure 2. The image in Figure 2a shows the scanned facility having 2900 m$^3$ with two stories and 418 m$^2$ of gross floor area. The opening step of the LS was to establish scan points. The image in Figure 2c demonstrates six scan points to achieve the point clouds of the facility, five positions around the facility, and one on the rooftop. The positions were prescribed to fortify a sufficient overlay of scanned areas to secure a concrete configuration of the facility. Light green lines around the five red circles in the picture in Figure 2c show the border line of the facility. The scanned results were then sophisticated through eliminating needless noise for reshaping the facility and merged together. The merging process was implemented along with the overlapped positions. On the basis of the merged shape, the outline and surface of the facility were generated as presented in Figure 2d–f.

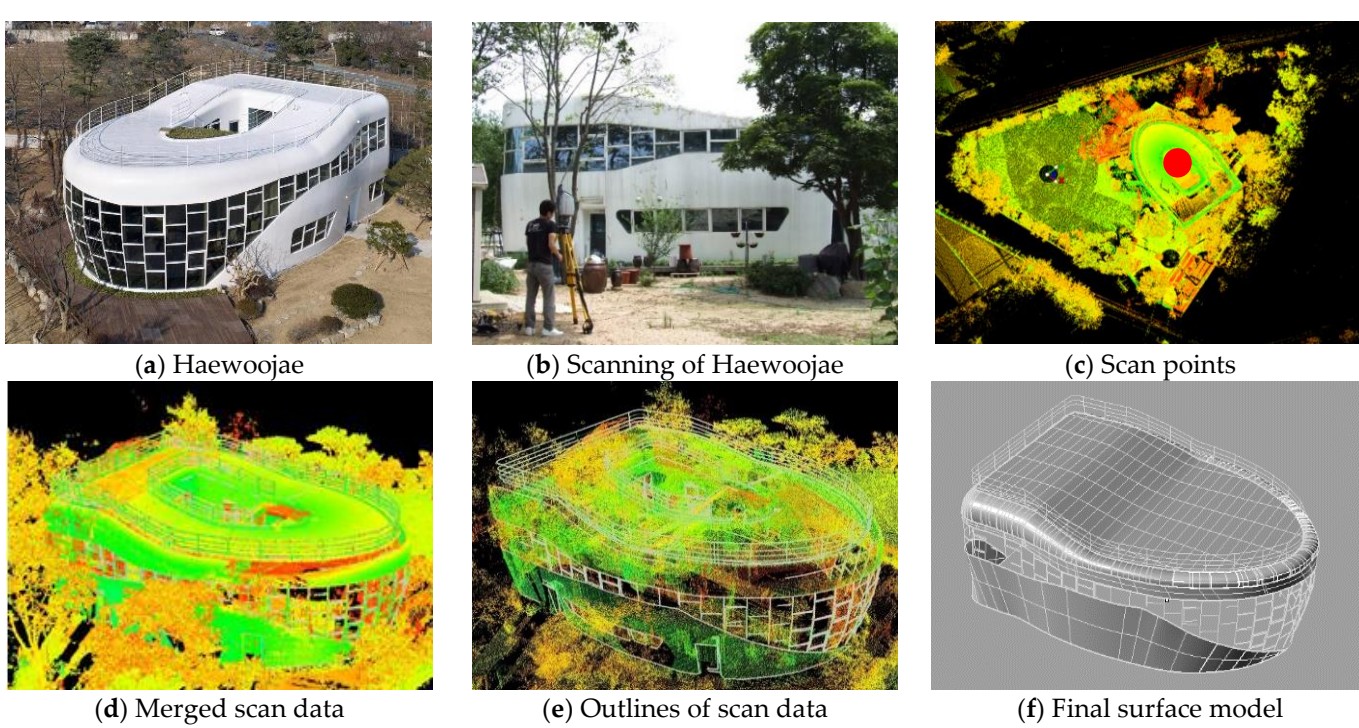

(**a**) Haewoojae     (**b**) Scanning of Haewoojae     (**c**) Scan points

(**d**) Merged scan data     (**e**) Outlines of scan data     (**f**) Final surface model

**Figure 2.** An example of reverse engineering based on the Scan-to-BIM process [14].

### 2.3. Drone Education

A drone has been extensively operated to collect data throughout diverse careers and research fields. This phenomenon has integrated drones into education in the engineering, construction, science, and technology professions. Al-Tahir et al. [28] suggested three methods to utilize drones for education: carry out a capstone design using a drone, combine a drone operation into a current training program, or develop a new program totally on a drone.

Bolick et al. [29] developed the drone education program which included lectures and lab exercises for the students of natural resource science. The students were trained on the procedures of acquiring and processing drone data through lectures and videos of drone simulation. Whilst the students expressed positive opinions overall about the drone education program, some of them stated their preference for hands-on experience rather

than simulation. Additionally, He et al. [30] devised a virtual drone training module to educate students how to obtain and process drone data into a 3D model reconstruction of an open-pit mine.

Although it is presumed that virtual drone education is beneficial for comprehending drone usage in construction, the best practice is definitely to perform in-person education. Sanson [31] introduced that the Civil and Construction Engineering Technology (CCET) Department of Youngstown State University (YSU), Ohio, USA, has taken actions to employ drones in their curriculum. Additionally, as mentioned earlier, the CONECTtech Lab at Georgia Tech, Georgia, USA, has recently carried out a new course called "Technology Applications in the Construction Industry [16]." The course encompasses multiple construction technologies, i.e., LS, 4D BIM, photogrammetry, Virtual Reality (VR), mobile applications, and project management software. However, the main concentration and the major attraction for students of the course is positioned on drones.

The construction industry in Korea faces difficulties in utilizing drones as well as laser scanners. The major challenging hurdle is to find skilled and knowledgeable manpower. While a number of Korean construction companies still hesitate to adopt digital technologies into their business having the innate conservatism of the construction industry and anxiety in mind owing to their unfamiliarity with the technologies, a number of construction professionals acknowledge digital technologies as an essential tool for business sustainability in the future. To stay competitive in the digitalized market trend, construction professionals should keep on upgrading themselves to be knowledgeable about applying digital technologies to their business.

*2.4. Drone and Sustainability*

The construction industry worldwide has been considered as a main donor to global carbon emissions, accounting for about 11% of global $CO_2$ gas emissions [32]. More than that, the built environment including buildings and social overhead capital promotes 40% of yearly energy consumption and 30% of all energy-related $CO_2$ gas emissions [33]. As such, the construction industry should perform a significant role in the transition to a zero-carbon world. Sustainable construction is the philosophy and the methodology in performing the role focusing on sustainability [34]. There are two directions that lead to sustainable construction: using recyclable and renewable materials and utilizing methods that can minimize energy spending and waste production.

Drones are one of the methods that help the construction industry become more sustainable. They can keep contractors regularly monitoring and inspecting construction progress and recognizing problems in real-time to input adjustments as necessary. This can support lesson rework and decrease the carbon footprint of construction projects. When work is conducted improperly in the projects, contractors should extend project schedules and frequently assign more cash and resources to rework. Actually, greater than 30% of total construction work on average in a project is connected to rework, squandering resources and increasing costs [35].

For the purpose of decreasing the ecological effect from fuel burning, contractors are able to utilize drone data to efficiently assign heavy equipment on the construction site. Since drones can precisely measure different areas for a project during the planning process, the contractors can prevent earthmoving equipment from unnecessarily moving back and forth across the jobsite or from waiting idle with engines being operated [36]. Additionally, drones can augment jobsite safety by finding risks and hazards even before the first day on the job. By combining this activity with the pre-planning process, the contractors can prepare to work around the risks.

Balasubramanian et al. [37] proposed an inclusive Construction 4.0 sustainability context that ascertains the Construction 4.0 technologies including a drone and their optimistic and pessimistic effects on social, environmental, and economic sustainability. The findings specified that the sustainability framework could be applied to the enhancement of the suitable positioning of digital technologies for sustainability.

## 3. Experimental Design

### 3.1. Course Description

As mentioned earlier in the research methodology section, a drone was applied to the FFE class. The class has been open several years before to provide students with procedures and theories of designing free-form facades based on the principles originated by Frank Gehry, the distinguished free-form building designer in the USA [38]. The class has utilized the LS technology to secure as-built quality of the free-form facades, conducted the data superimposition technique to measure discrepancy between as-built and as-designed free-form facades, and implemented the RE procedures in conjunction with LS application. The FFE is a three-credit undergraduate elective coursework and can be taken at the graduate level. The students in the course are in general architectural-engineering majors, and a majority of the enrollment is seniors with a few juniors and graduate students, typically having more or less 20 students.

### 3.2. Experimental Settings

Nineteen students, i.e., twelve seniors and seven part–time master-level graduates participated in the experiment. Table 2 represents the Construction Capacity Evaluation Criteria (CCEC) prepared by the MOLIT, Korea, in 2021. The CCEC, having been developed since 1996, groups Korean general contractors into 8 levels in accordance with their contract volume last year, technical competitiveness, and financial status [39]. Belonging to a higher level indicates that the contractors in the level can be awarded more volumes of works than the contractors in lower levels. The CCEC, however, does not describe how competent contractors are in effectively utilizing digital technologies. Among the 7 graduates, 4 have been working for the companies in level 1, and 3 others have been working in level 2. All of the 7 graduates have had more than 6 years of field experience as a quantity surveyor, scheduler, or field manager. As a matter of course, the number 7 is too small to represent reality of the digital innovation levels of Korean construction engineers. However, it could be a clue to infer that many Korean companies even in levels 1 and 2 are simply in infancy toward pursuing digital implementation using drones or laser scanners.

**Table 2.** Construction capacity evaluation criteria in Korea [39].

| Level | Construction Contract Volume Last Year (M: million) | Numbers | The Predesignated Scope of Project Sizes to Be Able to Contract (Before Bidding Price) | |
|---|---|---|---|---|
| | | | Civil Works | Architectural Works |
| 1 | More than USD 500 M | 58 (0.46%) | More than USD 142 M | More than USD 100 M |
| 2 | USD 500 M–USD 100 M | 137 (1.08%) | USD 142 M–USD 79 M | USD 100 M–USD 79 M |
| 3 | USD 100 M–USD 50 M | 176 (1.4%) | USD 79 M–USD 46 M | |
| 4 | USD 50 M–USD 27.5 M | 302 (2.4%) | USD 46 M–USD 33 M | |
| 5 | USD 27.5 M–USD 16.7 M | 500 (4%) | USD 33 M–USD 18 M | |
| 6 | USD 16.7 M–USD 10 M | 823 (6.5%) | USD 18 M–USD 11 M | |
| 7 | USD 10 M–USD 6.7 M | 640 (5.05%) | USD 11 M–USD 6.7 M | |
| Others | Less than USD 6.7 M | 10,015 (79.2%) | Less than USD 6.7 M | |
| Total | | 12,651 (100%) | | |

Two facilities were arranged for the experiment as shown in Table 3. Each of the facilities includes different shapes and complexity. Since the experiment was proceeded in the campus, the authors had to be very careful in selecting the facilities in order not to cause any incidents to other buildings as well as pedestrians and any trouble to other classes due to drone noise. The size and shape of the facilities were chosen to support the students to comprehend the workability level of a drone as well as a laser scanner in capturing as-built environments of the facilities in reflection of their configuration.

**Table 3.** The facilities used in the experiment.

|  | **Facility 1** | **Facility 2** |
|---|---|---|
| View | 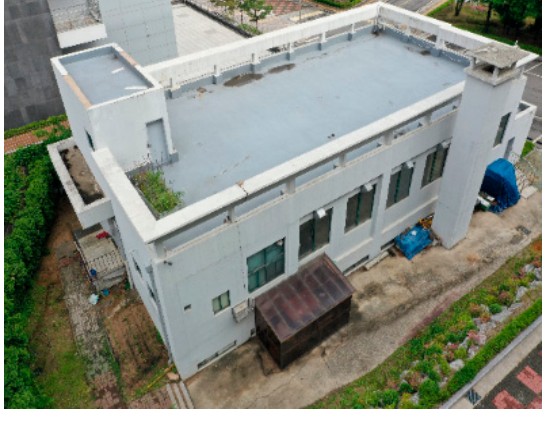 | 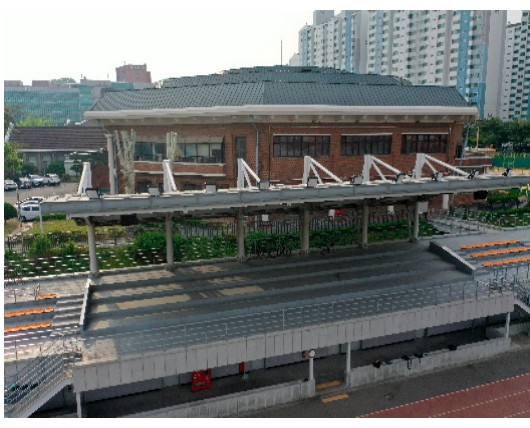 |
| Details | A power plant, one-story, 360 m², rectangular shape, no canopy, several projecting structural elements | A playground stand, 1060 m², rectangular shape, canopy roof |

The first facility in Table 3 is one story and a small power plant. It is located at the outskirts of the campus so that little interference and congestion by pedestrians was expected during the experiment. The exterior of the facility is finished with white paint, and the configuration of the parapet and the chimney imposes a little complexity on the facility. The second is a playground stand, located at the edge of the campus as well so that few restrictions seemed to be projected onto students' activities in the experiment. The exterior of the stand is constructed with steel pipes and panels. The canopy roof and the stand chairs seem to make it complex to gather the as-built data of the facility.

The authors organized the students into 6 groups, i.e., 3–4 students in a group. While it was desirable for each student to be involved in the experiment of the two facilities, therefore making it possible to obtain more experiment data, due to the time limitation of a semester, three groups of students were solicited to perform the experiment of a drone flight toward one building, e.g., groups 1, 2, and 3 to the power plant.

The experimental FFE class consisted of two modules: the lecture module and the experiment module. The lecture module progressed until the midterm covering the theories of designing free-form facades, securing as-built data of the facades with the drone and laser scanner and measuring the discrepancy between the as-built and as-designed categories. After the midterm, the students became involved in the experiment module for 7 weeks. The experiment covered six steps: (1) a theory and equipment-learning session, (2) a laser scanning operation, (3) laser-scanned data development, (4) a drone flight operation, (5) a drone-flown data exploitation, and (6) an experimental results presentation. Based on the experiment results, the students were requested to determine the pros and cons of the two technologies, the good or bad outcomes from the technologies in the experiment, and analyze the usability of the technologies in conjunction with the distinct features of the facilities.

### 3.3. Equipment

Table 4 shows the equipment as well as the related software used in the experiment. Among a number of laser scanners, this study utilized the Faro Focus 3D-X330 Time of Flight (TOF) scanner. A TOF laser scanner calculates the coordinates of its surrounding context in conjunction with the amount of time a laser signal needs to return to the scanner after reflected [40]. This procedure occurs a number of times until the designated range of data is achieved.

**Table 4.** The equipment used in the experiment.

| Equipment | Drone | | Laser Scanner |
|---|---|---|---|
| | DJI Mavic 2 Pro | DJI Phantom 4 Professional | Faro Focus 3D-X330 |
| |  |  |  |
| | Weight 0.9 kg, battery 30 min | Weight 1.3 kg, battery 25 min | Weight 5.2 kg, battery 4.5 h, measuring range 0.6–330 m |
| Software | Pix4D Mapper | | Revit Recap |
| | A photogrammetry software to transform the image from a drone to point cloud |  |  |

Two quadcopter drones were used in the experiment including a DJI Mavic 2 Pro and a DJI Phantom 4 professional. The former features a 3-axis gimbal with a 1* CMOS camera that provides 4K video, 20-megapixel photos, and filters [41]. It has a maximum flight speed of 72 km/hr and a maximum flight time of 31 min. The latter contains a 12.4 megapixel camera with 4K resolution video at up to 30 frames-per-second recording capabilities [42]. Its maximum flight time is up to around 28 min. The main drone prepared for the experiment was a DJI Phantom 4 professional. The DJI Mavic 2 Pro was adopted right after the Phantom 4 became unworkable due to a flight accident which happened in the middle of the experiment.

## 4. Experimental Process and Results

### 4.1. Flight Licenses and Permits

At the initiation of the experiment, the students were solicited to take a 6 h e-learning class regarding drones delivered by the Korean Transportation Safety Authority (KOTSA), a public entity to shelter lives and properties in all fields such as railroads, land, and aviation in Korea [43]. The contents of the e-learning class cover the fundamental knowledge of airspace system, particularly operation limits, UAV platform requirements, and the human operator's obligations.

The e-learning class is a mandatory qualification to fly drones with weight of 0.25–2 kg. To hover drones bigger than that size, operators should take a strict license exam and register the license. According to the Korean Aviation Safety Act (KASA), the license requirement is different with respect to the weight of drones ranging from the 1st to 4th levels: the 1st class license covering drones weighing 25–150 kg, the 2nd class covering 7–25 kg; the 3rd class covering 2–7 kg, and the 4th class covering 0.25–2 kg. While the first three level licenses require a certain period of aviation experience and an exam, the 4th level demands only taking a 6 h e-learning class without any aviation experience as well as an exam.

The experiment was not subject to any aviation approval from public authorities. As a matter of course, in Korea, when a person intends to operate a flight using a drone, they should obtain approval for the flight from the MOLIT by Article 127, KASA. However, the article provides clear exceptions for approval under the following circumstances: when the flying altitude is less than 150 m or when the weight on board including drone self-load is less than 25 kg. The drones adopted in the experiment were only 0.9–1.3 kg, and the flight height was less than 50 m.

### 4.2. Learning Session Proceeding

All the 19 students attended the learning session to clarify the operational process of the LS and drone flight and the activities to be executed in the experiment. In the first place, the scanner, a Faro Focus 3D-X330, was introduced to the students. Keeping the scanner on, the authors demonstrated not only how to set up the scanning parameters in the scanner to adjust scan resolution and quality, scan range, color setting, and scan duration but also how to reconstruct the as-built environment through refining and merging point clouds captured by the scanner.

As stated earlier, in order to secure safety during the experiment, two drone training professionals were invited from the private drone flight training institution which has been certified as a drone education provider in accordance with Article 126 (Designation, etc. of Training Center Specializing in Ultra-Light Vehicle), Korean Aviation Safety Act (KASA) [43]. The professionals delivered the presentation regarding the components of the two drones and their functions. They also demonstrated drone flights a few times by performing the following tasks: taking off and climbing to an altitude of 50 m, hovering in place, performing diverse flight patterns manually, showing automatic flying patterns, taking images of the facilities, and returning to the home location and landing.

### 4.3. Experiment Proceeding: Laser Scanning

#### 4.3.1. Setting up Scan Points

The first step of the LS was to establish scan points. Figure 3a,c presents seven and nine points founded to achieve the point clouds of facilities 1 and 2, respectively, maintaining enough intersection of scanned areas and thus gaining assertive shapes of the facilities. Each facility was scanned in the field just once by all the students together as a team. The individual groups of the student were then requested to refine point clouds by removing unnecessary noise in each scanned area and merge the points together to form a 3D model along with the overlapped location.

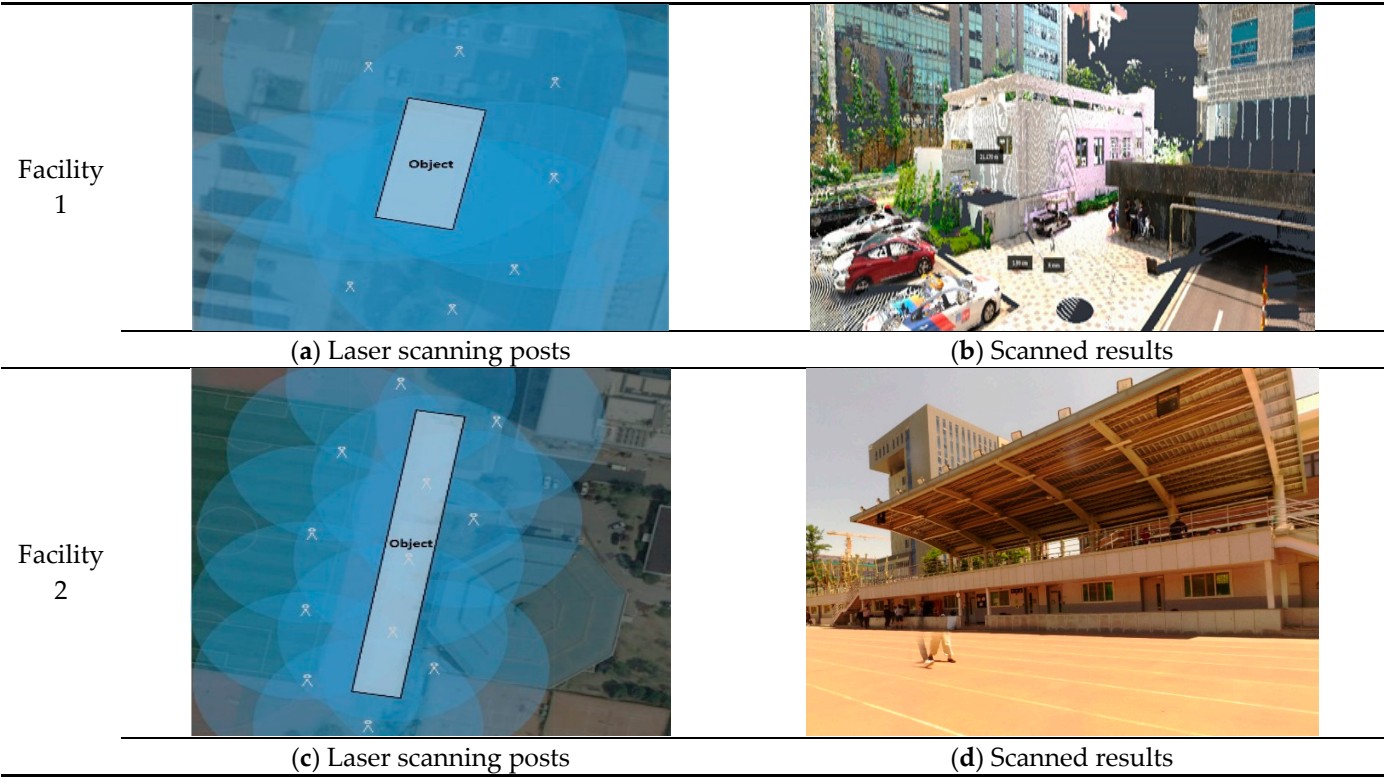

**Figure 3.** Results of LS application to facilities 1 and 2.

For facility 1, the LS was conducted for a total of 3.5 h at seven locations, maintaining a distance of 25 m from the center of the facility. In case of facility 2, the laser scanner placed four locations at a distance of 30 m from the center of the facility, two locations at a distance of 10 m from the side walls of the facility to scan the lower part of the canopy and stand seats, and three locations at a distance of 5 m from the backside wall of the facility to secure data in a narrow space between the stand and adjacent structures. A total of 6 h was spent in the LS of the facility.

### 4.3.2. LS Application Results

It is well known that LS allows contractors to provide both accurate and current data beneficial throughout the different design and construction stages, verify as-built models, and monitor progress on a project. Opposite to the usability, it also has certain weaknesses. The major disadvantage of laser scanners is the immobile nature of their collecting data [13]. Laser scanners assemble point clouds by spinning laser beams from a designated position and gauging the time taken for the laser to return to the scanner. As the laser tours only in a linear line, if any barrier is between the scanner and the object, a shadow is thrown that definitely triggers holes in the data.

To overcome this shortcoming, the scanner should be adjusted on site to make sure the particulars of the object are captured. Selecting suitable scanner positions is very labor-intensive and requires experience and skill. Even with the most skilled users, there are still surfaces, such as roofs, which cannot be scanned.

In the experiment, the students were requested to identify the disadvantageous circumstances that influenced the quality of the scanned images in addition to the drawbacks described above. They were also solicited to estimate if a drone could be an alternative to be able to overcome the disadvantageous situations. The following summarizes the findings from the LS application by the students and the critiques identified through the presentation after the experiment:

1. While clouds were generated in the experiment, the students indicated that since the main scanning locations were placed close to the parking lots of adjacent buildings as shown in Figure 3b, the frequent vehicle traffic and reflection lights from the vehicles were likely to hinder them from having high quality outputs from scanning.
2. The exterior of facility 1 is finished with white paint that reflects light. The bending in the point clouds of picture (b) in Figure 3 seemed to occur due to the light reflection. Previous research presents that strong light and the glare of the sun can limit the scanner's ability to capture desired objects [44].
3. The authors guided the students that whenever scanning a facility, the primary task was to organize the scanning job overall, which covered imagining the desired scan result, observing the areas to be scanned, and then simulating it to determine the best scan positions to achieve the result. Figure 4 shows the scanned outcomes of facility 2. These were the outcomes from the test scanning of the facility to select appropriate scan positions for the experiment. It is quite easy to acknowledge that the scan points were too far from the object, so the scanned results were not able to deliver what constructional elements were involved in the facility.
4. As stated above, 3.5 h were taken to scan facility 1, and 6 h were taken to scan facility 2. In cases of scanning buildings, properly estimating the amount of time of scanning operation in the field and the frequency of pedestrian traffic around the buildings allows for choosing the best time to perform scanning [44]. People and vehicles moving around the buildings can act as temporary obstacles that cause shadows and damage the quality of the scanning, frequently leading to rescanning. The two pictures on the lower side in Figure 4 demonstrated the shadows triggered by the people between the object and the scanner. The shadows lead to missing data and noise in point clouds, requiring a big investment in time and manual effort to remove them.

5. The students were able to appreciate the difference of scan resolution, i.e., high or low resolution, through comparing the scan result represented in picture (d) in Figure 3 with those in Figure 4. Resolution is the smallest possible distance between any two given points within a 3D model [45]. The higher the scan resolution, the better the details of the scanned data as well as more scan time. The students were instructed how to set the scan resolution parameters in the learning session of the experiment for efficient scanning operation.

6. Since the scanner was set on the ground, the scanning operation proceeded from the bottom to the top of an object using a bottom-up angle. Because of the bottom-up angle layout, certain parts of a building's exterior were not accessible to being captured in the scan data, such as a rooftop. Pictures in Figure 4 and picture (d) in Figure 3 did not deliver the overall roof shape of facility 2 due to this reason. The scanner should have been positioned on the rooftop of the stand to obtain the point clouds of the stand roof shape. There was no ladder or corridor that allowed for carrying the laser scanner up to the position. This constraint made it impossible to complete reconstructing of the overall 3D shape of the stand.

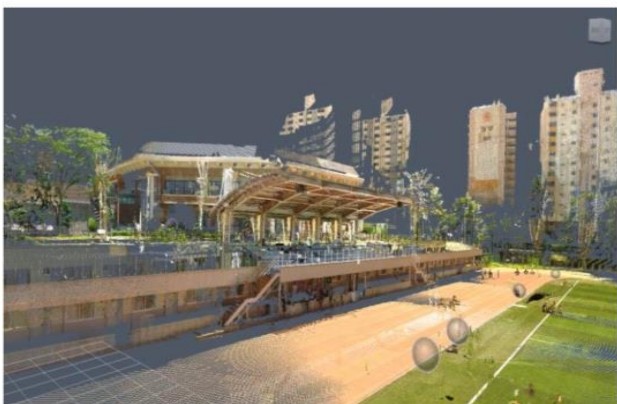
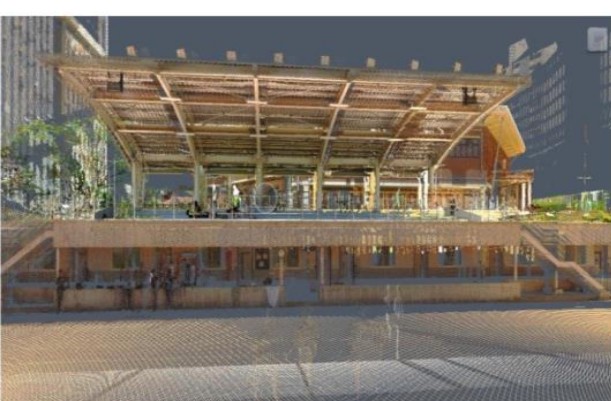
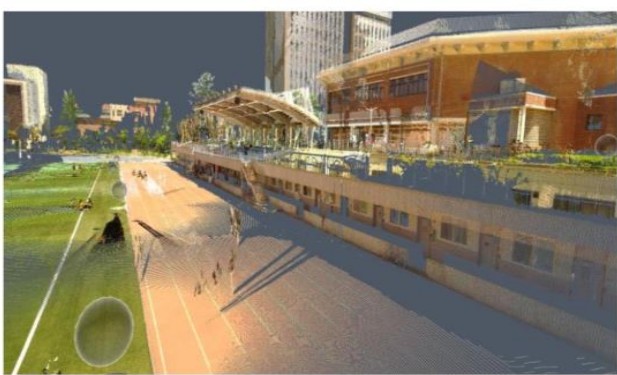
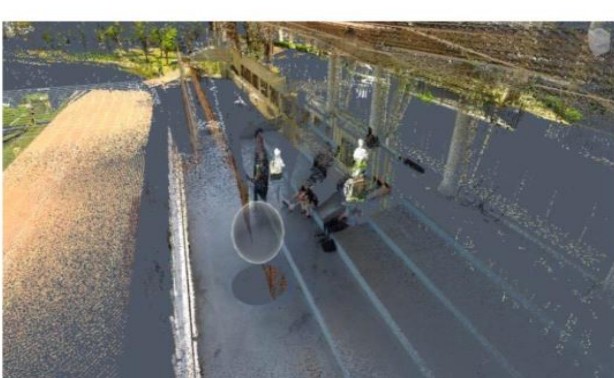

**Figure 4.** Results from the test scanning of facility 2.

*4.4. Experiment Proceeding: Drone Flight*

There are three types of drone operation control in accordance with the intervention level of a drone pilot: manual control, automatic control, and autonomous control [46]. Whilst autonomous control refers to conducting a drone flight without the interference of a pilot with the help of AI, automatic control indicates that a drone flies pre-set routes designated by a drone pilot before starting the flight. Manual control indicates that the pilot determines the drone's flight path. The authors guided the students to utilize manual control for the purpose of leading them to be intimate with drone operation.

### 4.4.1. Drone Flight Routes and an Accident

The blue dotted lines in the pictures of Figure 5a,b show the planned routes on which the drones were supposed to be aviated along the facilities. Each group of students flew the drones around one of the facilities assigned to them for around 25–30 min while taking photos. In the case of facility 1, operation was conducted to secure the overall data of the facility by maintaining a distance of 8 m to 25 m from the center of the facility. An average of 83 images was collected with an overlap of 50 to 60%. In the part where filming was disadvantageous due to adjacent facilities, 10 pictures of each end were additionally filmed with an overlap of 70%. The angle formed by the camera was maintained at 30 to 60°. In the case of facility 2, by maintaining a distance of 20 m and 60 m from the center of the facility, manual shooting collected an average of 224 images in 30 min with an overlap of 40 to 50%. The angle formed by the camera was maintained between 30 and 90°.

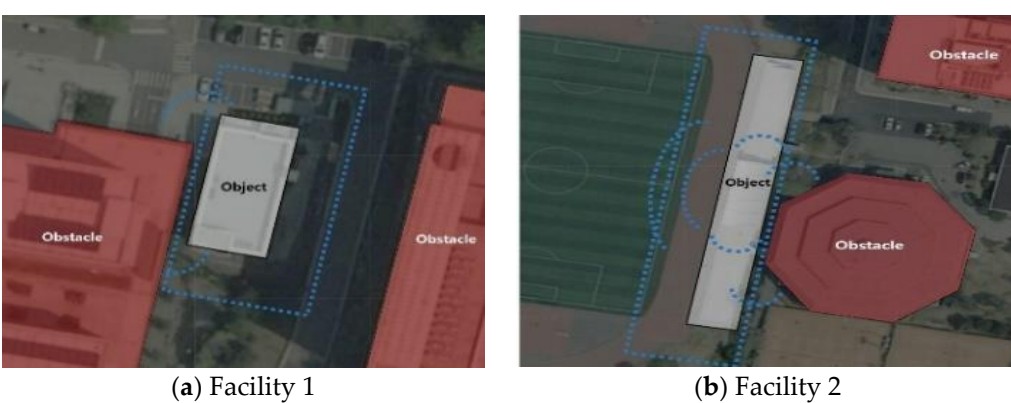

(**a**) Facility 1           (**b**) Facility 2

**Figure 5.** The drone aviation routes of facilities 1 and 2.

In the middle of the experiment with the DJI Phantom 4 professional, a drone crash accident happened around facility 1. As shown in Figures 4a and 3a, facility 1 was located very close to an adjacent building. The distance between facility 1 and the adjacent building is 5 m. According to the student behind the accident, when he/she moved the drone promptly back from the direction of the adjacent building to facility 1, the drone crashed against the side wall of the building and fell down to the ground. Fortunately, there were no pedestrians that morning. Three propellers were broken, but the students participating in the experiment stood away from the accident spot more than 10 m, so the fragments of the propellers did not hurt anyone.

A number of previous drone crush reports identify a variety of reasons for drone accidents such as malfunctioning rotors, no Global Positioning System (GPS) signal, compass errors, disconnected video transmission, an incorrect home point, power failure, insufficient battery, and pilot inexperience [47,48]. It was not clear why the accident occurred in this experiment. However, considering such comments by the students in the experiment that the drone did not show any problems before the accident, the cause of the accident seemed to be abrupt operation of the control sticks in the remote controller due to insufficient flight practice. The participants empirically realized that although the drone is equipped with an obstacle sensing system that can identify obstacles in front of and below it to support it to avoid clashes, the system was not a panacea to prevent an accident.

### 4.4.2. Drone Application Results

Figure 6 shows the transformed meshed shapes of facility 1 and 2, respectively, from the point clouds with the photogrammetry software, Pix4D mapper, based on the number of pictures taken by each group of students with the drones. Pix4D mapper is a drone mapping software that works with drones to transform a large number of images into accurate point clouds and 3D textured mesh [49]. The software that provides a similar function to Pix4D mapper includes PhotoScan, DroneDeploy, and Precision Hawk.

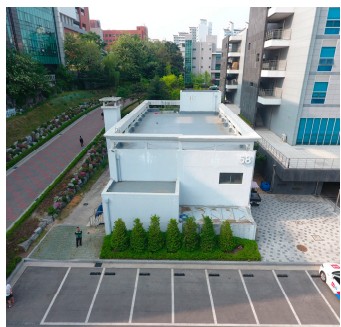

(**a**) Image from drone

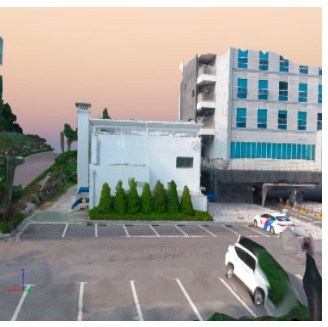

(**b**) Transformed meshed shape with 56 pictures

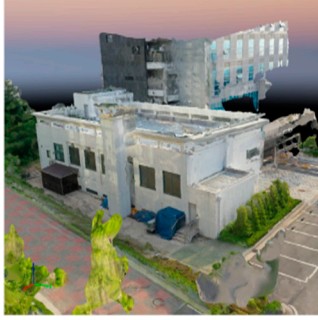

(**c**) Transformed meshed shape with 87 pictures

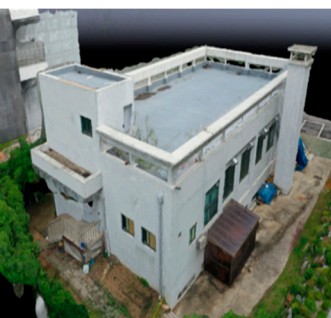

(**d**) Transformed meshed shape with 53 pictures

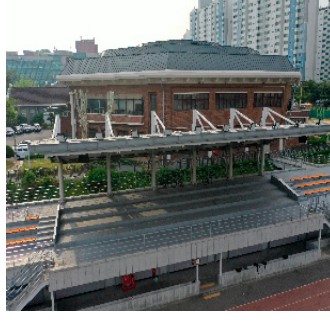

(**e**) Image from drone

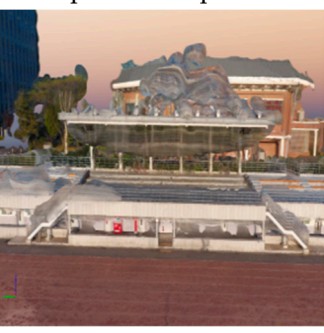

(**f**) Transformed meshed shape with 168 pictures

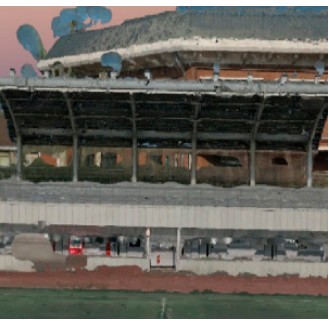

(**g**) Transformed meshed shape with 352 pictures

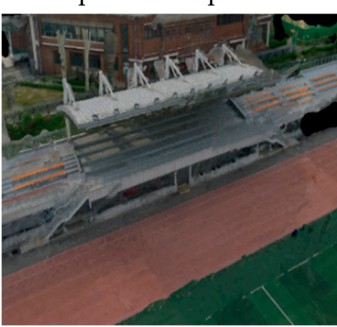

(**h**) Transformed meshed shape with 83 pictures

**Figure 6.** Results of drone application onto facilities 1 and 2.

Reviewing 3D modeling results in Figure 6, i.e., pictures (b), (c), and (d) of facility 1 and pictures (f), (g), and (h) of facility 2, the shapes modeled with Pix4D mapper were much too vague to determine the configuration of the facilities compared to the images taken by the drones, i.e., pictures (a) and (e). The overall quality of the experiment results was worse than expected and acknowledged not to be applicable to the procedure of developing a UAV-to-BIM application without certain remedy actions to improve the quality. Whilst it is necessary to analyze and research the reasons for those elusively transformed outputs, it seems to be obvious that the time given to the students was not enough to reconstruct the images into a 3D model from the drone beginners' point of view. A number of drone professionals emphasize that operating a drone is more about the data and presentation than about the flying. The study confirmed the expression and the need for advanced education to comprehensively plan a drone flight and skillfully handle the outputs from the flight.

Whereas 30 min was taken for each group of the students to photograph the individual facility, manipulating the photo images with the Pix4D mapper took 3 h on average. After photographing the facilities, the students followed the modeling procedures presented in the Pix4D mapper manual to create 3D models of the facilities consisting of initial processing, densifying point clouds, and delivering 3D textured meshes [49]. The following summarizes the findings from the drone experiment by the students and the critiques delivered through the presentation after the experiment.

The students felt at first nervous about flying a drone but found that it was very much easier than expected as soon as the drone started to hover. They were inclined to move it faster and faster similar to a toy in spite of being educated in the learning session to fly it as slow as possible around the facilities to prevent any incidents from happening.

1.  The drone education professionals conveyed that safety should be a primary concern when dealing with a drone; however, during the experiment, safety around the experimental surroundings did not appear to be the first issue to be met. The learning

session was too short for the students to understand that a drone was an aircraft system that should be managed methodically and strictly for safety as a unit including a pilot on the ground, the drone itself, a control system, and communication links.

2. Reviewing the accident case, it was very clear that if a building to be photographed with a drone was very adjacent to a nearby building, special caution was required to avoid disastrous incidents. Additionally, a specific image acquisition plan was necessary for accurate and sharp images reflecting the narrow configurations of the buildings.

3. In pictures (f) and (g) in Figure 6, it was easy to recognize noise above the roof of the stand. The students who worked on the facilities indicated that this could be caused by the rolling shutter phenomenon during the drone rotation process when photographing. It was discussed in the presentation session that the indication was not reasonable because before operating the drones, the shutter speed, aperture, and ISO in the cameras of the drones were configured on automatic settings so that the shutter control hardly influenced the quality of the images taken.

4. Some students questioned whether the noise in pictures (f) and (g) could be due to sunlight. They wondered that if building facades were finished with the materials that reflected sunlight as described in the laser scanning experiment, errors in obtaining images were likely to occur. The two professionals explained that while t might be one of the causes of the blurry modeling results, a more realistic cause would be improperly acquiring the images. When obtaining the images, the first item to be satisfied was the high overlap between images since occupying the image dataset to create point clouds relies on visual similarities between the overlapped images.

5. Although the overlapping of images was significant, it was questionable for the students to be able to maintain a suitable degree of overlap during the experiment since they were inexperienced in flying a drone and, even worse, requested to use manual flight mode in spite of their inexperience. While more specific guides were available from the manual for securing enough overlap of the images such as "fly around the buiding a first time with a 45° camera angle; fly a second and third time increasing the flight height and decreasing the camera angle with each round; take one image every 5 to 10°depending on the size of the building and distance to it," the students should have practiced a lot to satisfactorily follow the designations in reality.

6. Pictures (b), (c), (d), (f), (g), and (h) in Figure 6 are the representation of the 3D textured mesh delivered from the densified point clouds. Viewing pictures (c) and (h), in addition to the noise around the facilities, the roof and parapet shapes of facility 1 and the corridor and roof shapes of facility 2 were very undistinguishable and inaccurate. The mesh receives the point clouds as input. So, if the point clouds are noisy, the mesh also noisy. If the quality of the images is not high, the 3D textured mesh generated shows successively a tendency to have holes or not be planar in planar surfaces due to the low quality of the point clouds.

7. Picture (d) in Figure 6 was much clearer than any other pictures. The students in the group that worked on picture (d) indicated they operated the processing options provided in the Pix4D mapper to edit and filter the noise in the point clouds and to improve the quality of 3D textured mesh, e.g., the noise filter processing option to furnish cleaner point clouds for datasets and the sky filter processing option to remove points in the dense point clouds associated with sky.

8. On the contrary to the laser scanner, the drones could easily capture the rooftop images of the stand by means of a top-down angle. However, when taking images under the roof of the stand shown in pictures (f) and (g), a certain limitation was found caused by a gimbal configuration of the drones. The three-axis gimbal furnishes a stable platform for the attached camera, tolerating for smoothly moving and capturing clear images. The gimbal installed in the drones could tilt the camera within a 120° range consisting of up to 30° upward and 90° downward from the horizontal line. However, due to the upward sloping limit of the gimbal, it was very difficult to capture the

images in the ranges bigger than the limit. The difficulty led to shadowy images causing blurry shapes of the 3D textured mesh.

## 5. Recommendations

The authors conducted an FGI-based workshop with the two drone education professionals and seven graduates who participated in the experiment to identify the recommendations for developing the drone curriculum for the construction engineers. In the process of FGI, a moderator asks a question to the participants and allows natural conversation to rise with respect to the question. The authors played the role of the moderator and identified valuable recommendations with the interviewees together. The nine participants were solicited to describe their opinion on the following questions: (1) Is it necessary for developing the drone education curriculum in the universities for construction engineers? (2) What contents are essentially embraced in the curriculum? (3) What formats and supports are desirable to successfully develop and implement the curriculum? The following are the recommendations acknowledged from the workshop.

### 5.1. The Need for the Curriculum

All the workshop participants agreed that the formal drone curriculum should be prepared the sooner the better for the construction engineers to keep up with the current digitalized market trend. While both a drone and a laser scanner allow for digitally restructuring accurate as-built models, the technologies include distinctive pros and cons, respectively, as shown in the experiment so that there is a need to go hand in hand to successfully function construction quality monitoring. Compared to LS, drone application is still at an infancy level in all life-cycle stages of built environments in Korea. Education on how to collaborate a drone with a laser scanner will support construction engineers to be confident and insightful in challenging construction operation.

As stated earlier, the majority of the engineers in the Korean construction industry are novices at utilizing a drone in construction operation. While common drone training programs are available from the current market place, those are simply to deliver how to fly drones without covering the distinctive characteristics of construction operation. Construction activities are very systematic in the sequential processes of planning, design, construction, and maintenance. The BIM data generated in the former process should be used in the next process and applicable to meet very long-term sustainability of a construction output. The drone curriculum in conjunction with the CE&M context is expected to support the engineers to enhance productivity of their construction activities.

### 5.2. The Contents of the Curriculum

The participants came to understand that hovering a drone was 20% of the drone job and the other 80% would be data processing. As mentioned above, the drone training programs in the market are not to deliver the suitable capacity of managing the drone data such as point clouds, 3D textured mesh, and others but simply to instruct how to fly a drone and obtain a license. The participants emphasized that the contents of the curriculum should focus on acquiring, processing, analyzing, and improving the drone data rather than on flying a drone. There are diverse commercial applications to handle the digital data captured with a drone such as Pix4D Mapper, PhotoScan, DroneDeploy, Precision Hawk, and 3D Robotics. The curriculum needs to determine and deliver the common procedures and techniques involved in each of the applications.

The participants emphasized that the contents of the curriculum also needed to deal with drone risk. Flying a drone is very challenging, especially when people are first involved in drone operation. As shown in the experiment, accidents are too easy to happen. Drone risk embraces physical as well as non-physical threats including safety, privacy, intellectual property, and operational security [50]. The curriculum must include information on how to establish safety protocols in the use of a drone. Rather than a manual flight, the use of best practices was insisted upon with the automation flight.

As shown in the experiment, even beginners in drone flight easily tend to regard drones as a toy and want to fly them faster. Time, practice, and experience are required to be a skillful operator. Very few studies have explored the human aspects of drone flight, i.e., drone operators' reasoning capacity and job performance [51]. All the workshop members agreed that the curriculum should address human factors and human performance in drone flight in combination with safety, experience, and a risk management plan. Human factors indicate the variables that influence a human's capability such as external, e.g., light or noise, or internal, e.g., fatigue, factors. Human performance, which is a function of human factors, denotes the human competence to positively achieve tasks [52].

The participants added that considering that construction projects were operated anywhere in the nation, the curriculum should educate the construction engineers to understand the drone regulations such as prohibited areas of the drone flight and administrative procedures to obtain flight permission in diverse regions. In Korea, according to the Radio Waves and Aviation Act, drone flight is banned within a 5.5 km radius of an airport, at an altitude of 150 m or higher, near the cease-fire line, over the city of Seoul, in no-fly zones, and also over densely populated or over-crowded areas. Additionally, there is different administrative complexity for obtaining a drone flight approval depending on region by region. In construction sites, the needs for drone operation occur irregularly and frequently. The construction engineers, therefore, need to know the general procedures to obtain the approval while complying with the regulations in different jurisdictions.

*5.3. The Desirable Formats of the Curriculum*

There are generally two types of curriculum models: the product model and the process model [53]. The product model focuses on outcomes of the curriculum while the process model centers on how learning is developed over time. The workshop participants discussed that since the drone curriculum is aimed at generating very clear digital outcomes by teaching the specific step-by-step procedures, the product model would be a proper setting to pursue. Additionally, among three curriculum design methodologies widely accepted in the field of curriculum development, i.e., the subject-centered, problem-centered, and learner-centered approaches, the subject-centered approach would be appropriate because the curriculum was supposed to be a basic one upon which the problem-centered and learner-centered ones could be developed at the graduate study level.

The two drone professionals suggested that since flying a drone in the campus was very risky due to lots of students, traffic, and buildings, dividing the curriculum into two modules, i.e., a drone flight module and a data processing module, could be a reasonable approach to manage the risk. Through industry–academia collaboration, the drone flight module could be then outsourced to private institutions. Doing so, faculties could focus more on teaching the disciplines of processing drone data while mitigating the drone operation risk on campus and the burden of hiring a qualified pilot for the curriculum.

Instead of dividing the curriculum, a different option was discussed, that is, to require students for a certain prerequisite to register for the drone class. The prerequisite could be obtaining a drone license from private drone schools or taking on-line training courses. Some workshop participants were, however, skeptical of this option, pointing out that it would not be workable because the option imposed an extra load onto the students. They insisted that before inventing the option, the suitable format of the curriculum to the entire CE&M program, i.e., a core course or elective course, the duration of the curriculum, and the correlation of the curriculum with other courses in the program, should be contemplated in the first place. Among the three different curriculum formats to introduce drones into education [28], the authors targeted the development of the curriculum as an elective course having the format of incorporating drone education into an existing course in combination with LS technology during one semester.

Developing the curriculum requires the school to provide diverse resources to purchase drones and insurance, outsource drone flight education, or hire full-time or part-time drone pilots. It was recommended that the school administrative workforce should be

consulted to secure the budget for the items, thereby making the curriculum realistically reasonable.

*5.4. The Preliminary Curriculum Framework Recommended*

Based on the experiment results and recommendations above, the workshop identified the preliminary curriculum framework shown in Table 5. As mentioned earlier, this research is a preliminary study to identify the necessary education for drone beginners to be knowledgeable about operating a drone and obtaining high quality modeling results.

**Table 5.** A Preliminary Drone Curriculum Framework.

| Learning Goals | Learning Objectives (●) and Activities (-) |
|---|---|
| Understand a drone as a system that can capture existing surroundings with a camera | ● Understand drone hardware, firmware, network systems, and their functions.<br>  - Recognize the different parts of a drone and their function.<br>  - Study basic troubleshooting for common drone problems.<br>● Understand basic drone movement and how to use a camera with a drone.<br>● Understand safety precautions when using drones.<br>● Learn how to aviate drones at manual, automatic, and autonomous modes.<br>  - Practice hands-on hovering and landing and video simulation exercises. |
| Understand the application of LS to monitoring the as-built construction quality of a facility | ● Learn the LS technique to capture as-built conditions of a facility.<br>  - Practice the Scan-to-BIM process through a simple hands-on case study.<br>  - Identify the influencing factors on the scanned results.<br>● Understand the BIM data superimposition technique to measure the difference of the as-built from the as-planned.<br>  - Practice data superimposition through a simple hands-on case study.<br>● Understand the pros and cons of the LS technique. |
| Understand the application of a drone to monitoring the as-built construction quality of a facility | ● Understand the effect on camera images from the interaction of diverse camera settings with natural conditions as well as image overlapping ratios.<br>  - Analyze the image quality deviations depending on various interactions.<br>● Understand how to make a reasonable image acquisition plan.<br>● Understand the data processing steps for obtaining high quality products.<br>  - Study the functions installed in diverse commercial applications.<br>● Understand the methods to improve the cons of LS as well as of a drone. |
| Understand the restrictions, risks, and human performance involved in drone operation in construction projects | ● Understand the legal requirements for flying drones in various project sites.<br>  - Investigate flight permit rules in diverse regional jurisdictions.<br>● Understand the effectiveness of a drone to enhance sustainable construction.<br>● Recognize the significance of safety concerns in drone aviation.<br>  - Identify crash accident cases; investigate the reasons and results of the cases.<br>● Comprehend the risks involved in drone operation.<br>  - Develop a risk management plan for educational purposes.<br>● Understand the significance of securing privacy from drone aviation.<br>● Understand human factors and performance in drone flight. |

The learning goals in Table 5 indicate the focuses to satisfy the expression of "knowledgeable about operating a drone and obtaining high quality modeling results" through education. On the contrary, learning objectives represent the specific and measurable sub-goals and inform particular learning outcomes. The learning objectives can then be

broken down into small learning activities which stand for knowledge or skills that are necessary for meeting the relevant learning objective.

## 6. Discussion

The limitation of the study is that the recommendations were identified on the basis of the opinion and discussion of the nine workshop participants. While it is expected that the findings from the study are reasonable and beneficial for the CE&M faculties to refer to when preparing a drone curriculum in universities, more comprehensive industry-wide opinions, possible barriers, and suggestions to overcome the barriers are needed in future research to establish the rationale for a further inclusive drone curriculum for construction engineers.

Apart from the research on the curriculum development, it is necessary for the authors to furthermore investigate the limitations that hinder spreading drone utilization in construction projects through more practice on drone application toward diverse facility types thereby definitizing effective directions and specific methodologies to surmount the limitations and keeping on upgrading the construction engineers.

Since this study focused on performing experiments with drone beginners for investigating their attitude regarding drone flight, it contained descriptive and behavioral findings rather than analytic. Additionally, this study was very exploratory in that there has been rarely such an empirical study in cooperation with drone beginners before. In the process of delivering the descriptive and exploratory research characteristics, this study addressed little about mathematical modeling of drone flight although it helps the drone beginners to understand drone behavior control.

Drones belong to a class of nonlinear systems that are inherently difficult to control [54]. Mathematical modeling is the first step towards comprehending the system dynamics and monitoring the response involved in drone behavior [55]. A number of distinguished drone researchers have presented various mathematical models that explain operational system analysis and controller design of drone systems [56,57]. Others have identified the beneficial methodologies with respect to drone flight schedules and flight trajectory, which enhance drone safety in the long run [58,59]. While the present study has not forwarded the diverse mathematical models, it is necessary for the authors to further develop an introductory curriculum that heightens the understanding of the mathematical models in future research for construction engineers.

## 7. Conclusions

The construction industry has been considered a key industry that is devoted to the economic development of a nation. Construction professionals recognize that digitalization cannot be avoidable and digital technologies are an essential tool for survival. Among diverse digital technologies, drones have been recently well acknowledged as a significant technique for collecting valuable data and insights in diverse construction processes. The MOLIT, Korea, announced a plan to utilize drones and LS for monitoring construction operation quality in major public projects by 2025. Whereas contractors need to train their manpower, the two technologies are not so popular in construction projects in Korea, and very few experts are available.

The Korean universities implementing CE&M were requested to develop a curriculum for providing education on the technologies for the contractors. Developing a curriculum is a multi-step process of creating and improving a learning objective and teaching strategies, materials, and assessment. To avoid trial and error in progressing the drone and LS curriculum and successfully implementing it later on, it is very essential to perform a preliminary study to identify the needs for the curriculum. The objective of this study is to carry out a preliminary study through an experiment with drone and LS applications for beginners in the construction industry and to identify valuable lessons that would be beneficial for promoting the curriculum.

The authors arranged 19 students, 12 seniors in the undergraduate program and 7 part-time graduate students in the CE&M program at SNUST in Korea, as the participants of the experiment. Two facilities with different shapes and sizes on campus were selected as the subject facilities to be experimented, i.e., a 360 m$^2$ power plant with one story and a 1600 m$^2$ playground stand. The authors incorporated a drone education module into one of the existing CE&M classes, called FFE in the SNUST. Two drone education professionals were invited from a private drone flight training institution to help with the experiment.

In the experiment, a Faro Focus 3D-X330 TOF scanner as well as two quadcopter drones including a DJI Mavic 2 Pro and a DJI Phantom 4 professional were utilized. The experiment consisted of six steps: (1) a theory and equipment-learning session, (2) a laser scanning operation, (3) laser-scanned data development, (4) a drone flight operation, (5) drone-flown data exploitation, and (6) an experimental results presentation. Based on the experiment results, the experiment participants determined the pros and cons of the two technologies, the good or bad outcomes from the technologies in the experiment, and analyzed the usability of the technologies in conjunction with the distinct features of the facilities.

After the experiment, the authors conducted an FGI-based workshop with the two drone education professionals and seven graduates who participated in the experiment. In the workshop, the nine participants were solicited to describe their opinion on the needs for the curriculum, the contents of the curriculum, and the desirable formats of the curriculum as the recommendations from the study that are beneficial to develop the comprehensive drone education curriculum later on.

The recommendations include: drones and laser scanners need to go hand in hand to successfully function construction quality monitoring; the contents of the curriculum should focus on processing drone data rather than on flying a drone; there is also a need to address drone risk and human factors as well as human performance in drone flight in combination with safety, experience, and a risk management plan; the product model and subject-centered curriculum design are suitable to the curriculum, separating a drone flight module from a data processing module in the curriculum, and utilizing industry–academia collaboration can be a reasonable approach to manage the drone risk.

**Author Contributions:** Conceptualization, J.O.; methodology, J.O.; validation, J.O.; formal analysis, J.O. and S.M.; resources, S.M.; data curation, S.M.; writing—original draft preparation, S.M.; writing—review and editing, J.O.; visualization, S.M.; supervision, J.O.; project administration, S.M. All authors have read and agreed to the published version of the manuscript.

**Funding:** This study was supported by the Research Program by the SeoulTech (Seoul National University of Science and Technology), Korea.

**Acknowledgments:** This study was supported by the Research Program funded by the SeoulTech (Seoul National University of Science and Technology), Korea.

**Conflicts of Interest:** The authors declare no conflict of interest.

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
