# Peer review of "Developing the Framework of Drone Curriculum to Educate the Drone Beginners in the Korean Construction Industry"

_drones, doi:10.3390/drones7060356_

Round 1
Reviewer 1 Report
please consider the critics

see the critics
Author Response
Dear Reviewer,
Thank you so much for your considerate review of the manuscipt. The authors reflected your comments into the manuscipt. Please refer to the file attached. Thank you.

Reviewer 2 Report
No comments.
Author Response
Dear Reviewer,
Thank you very much for your considerate review of my manuscript.
Reviewer 3 Report
In this work titled Developing the Framework of Drone Curriculum to Educate 2 the Drone Beginners in the Korean Construction Industry the authors dealt with Drone, Laser Scanning, Drone Curriculum, Construction Operation Monitoring, Smart 25 Construction, Construction 4.0
The paper is presented according to the topics : Introduction, Literature Review, Experimental Design , Experimental Process , Recommendations, Discussion ,nd Resultsa nd conclusion and list of the references.
In this paper, the authors donde some “ recommendations include: drones and laser scanners need to go hand in hand to successfully function construction quality monitoring; the contents of the curriculum should focus on processing drone data rather than on flying a drone; also need to address drone risk and human factors as well as human performance in drone flight in combination with safety, experience, and a risk management plan; the product model and subject- 794 centered curriculum design are suitable to the curriculum, separating a drone flight mod ule from a data processing module in the curriculum and utilizing industry-academia col- 796 laboration can be a reasonable approach to manage the drone risk” .
The paper is well written and with no mistakes and the authors addressed the main question posed .
The arguments presented by the authors were consistent with the evidence and arguments presented by the authors .
The concluding remarks were supported by the data.
We note that the authors mention only 53 references. Is it enough to a complete analysis of the state of the art ?
I don’t see any see any state of the art of mathematical modeling of an unmanned aerial vehicle?
Author Response
Please check for the response file attached.

Round 2
Reviewer 1 Report
it is ok